# Microsurgical Management of Male Infertility: Compelling Evidence That Collaboration with Qualified Male Reproductive Urologists Enhances Assisted Reproductive Technology (ART) Outcomes

**DOI:** 10.3390/jcm11154593

**Published:** 2022-08-06

**Authors:** Jessica Marinaro, Marc Goldstein

**Affiliations:** 1Department of Urology, Weill Cornell Medicine, New York, NY 10065, USA; 2Center for Male Reproductive Medicine and Microsurgery, Weill Cornell Medicine, 525 East 68th St., Starr Pavilion, 9th Floor (Starr 900), New York, NY 10065, USA

**Keywords:** assisted reproductive technology, infertility, microsurgery, urology

## Abstract

A male factor plays a significant role in a couple’s reproductive success. Today, advances in reproductive technology, such as intracytoplasmic sperm injection (ICSI), have allowed it to be possible for just a single sperm to fertilize an egg, thus, overcoming many of the traditional barriers to male fertility, such as a low sperm count, impaired motility, and abnormal morphology. Given these advances in reproductive technology, it has been questioned whether a reproductive urologist is needed for the evaluation and treatment of infertile and subfertile men. In this review, we aim to provide compelling evidence that collaboration between reproductive endocrinologists and reproductive urologists is essential for optimizing a couple’s fertility outcomes, as well as for improving the health of infertile men and providing cost-effective care.

## 1. Introduction

Infertility is defined as the inability to conceive after 12 months of regular, unprotected intercourse [1]. It is estimated that approximately 15% of all couples are affected by infertility [2]. A male factor is solely responsible in approximately 20% of couples, and contributory in another 30–40% [2]. While a male factor plays a significant role in a couple’s reproductive success, recent evidence suggests that many men with infertility are never referred to a reproductive urologist (RU) for evaluation [3,4].

While the reasons for this are likely multifactorial and have not been fully elucidated, one contributing factor may be a lack of referrals from reproductive endocrinologists (REs). A recent study by Samplaski et al. demonstrated that reproductive endocrinologists serve as the “gatekeepers” for male infertility referrals, with approximately 60% of the men seen by reproductive urologists being referred by an RE or women’s fertility specialist [3]. Among these REs, there is significant variation in the rates of referrals to reproductive urologists, ranging from 3.7% to 35.8% [4].

This variation in referrals from REs to RUs may be because assisted reproductive technologies (ARTs) are able to overcome many of the traditional barriers to fertilization associated with male infertility. Specifically, with the advent of intracytoplasmic sperm injection (ICSI), only one sperm is required to fertilize an oocyte, thus, overcoming the fertilization difficulties traditionally caused by low numbers of sperm, impaired motility, and abnormal morphology [5]. With these new ART techniques and high ART success rates, many men are never referred to a reproductive urologist for a complete evaluation.

In this article, we summarize the current literature available regarding the importance of evaluating infertile men, as well as the various surgical treatments that reproductive urologists can offer infertile couples. Our main aim and objective is to prove that a male evaluation is meaningful and important, not only for the couple’s reproductive success, but also for the overall health of the male partner.

## 2. Why Evaluate the Male?

### 2.1. Optimizing Birth Outcomes Involves More Than Just Performing ICSI

ICSI was initially developed to overcome the most severe forms of male factor infertility; however, in recent years, it has become the most common method of fertilization used for ART [6]. While the percentage of diagnoses of infertility due to a male factor has remained stable, ICSI use has steadily increased [6]. In the United States specifically, the use of ICSI for all indications increased from 36.4% in 1996 to 76.2% in 2012, with the largest relative increase seen among cycles without male factor infertility (15.4% to 66.9%, *p* < 0.001).

Given the widespread use of ICSI and its invasive nature, researchers have questioned whether this reproductive technique has adverse health consequences on offspring [7]. While several large meta-analyses have found an increased risk of congenital malformations among ART offspring compared to those conceived naturally [8,9,10,11], few studies have compared the risk of malformations among ICSI offspring versus those conceived with conventional in vitro fertilization (IVF). While there is limited research on this topic, there is some evidence to suggest that ICSI may be more hazardous to offspring than other reproductive techniques. For example, in one large observational study of over 300,000 births in Australia, only ICSI was associated with a higher risk of birth defects after a multivariate adjustment (OR 1.57, 95% CI 1.30–1.90); conventional IVF was not associated with any increased risk of birth defects after the same analysis (OR 1.07, 95% CI 0.90–1.26) [12]. Similarly, in a multicenter European cohort study of over 1500 children, Bonduelle et al. found that only those conceived through ICSI had a higher rate of major congenital malformations (OR 2.77, 95% CI 1.41–5.46); those conceived through conventional IVF did not have a higher rate of major malformations (OR 1.80, 95% CI 0.85–3.81) [13]. After adjusting for various sociodemographic and environmental factors, this higher risk of major malformations persisted for the ICSI children (OR 2.54, 95% CI 1.13–5.71) [13].

There is debate about whether this increase in birth defects is secondary to the ICSI technique itself or the underlying male infertility factors that necessitate its use. However, in the previously mentioned study by Bonduelle et al., oligospermia (defined as a sperm concentration < 20 million/mL) did not influence the presence of major or minor congenital malformations, lending support to the hypothesis that the ICSI technique itself may contribute to these birth defects [13]. Similarly, other studies of children conceived using ICSI performed for nonmale factor infertility indications found that ICSI in this setting was associated with a lower birthweight [14] and increased risk of autism (adjusted HR 1.57, 95% CI 1.18–2.09) [15] versus conventional IVF, again suggesting that the ICSI technique itself may play a role in the overall health of offspring.

Ultimately, the American Society for Reproductive Medicine (ASRM) concluded in a recent practice committee opinion that while ICSI has been associated with a small increased risk of adverse outcomes in offspring, this has generally been attributed to underlying male factor infertility [16]. However, they conceded that it is “unknown how these risks may be related to ICSI for non-male factor infertility patients” [16] and, additionally, high-quality research is needed to fully understand the role that ICSI itself may play in the overall health of offspring.

Until these risks are fully elucidated, there is an opportunity for reproductive urologists to collaborate with female infertility providers to optimize male fertility to achieve the best birth outcomes for the couple. Specifically, by treating male factor infertility, it may be possible to use less invasive reproductive techniques, which may lead to the delivery of healthier children.

### 2.2. Male Fertility Is Increasingly Associated with Overall Health

In recent decades, it has been increasingly recognized that male reproductive health and overall health are related [17]. Specifically, many of the conditions known to cause male infertility have also been associated with broader health consequences. One example is Klinefelter syndrome (KS, 47XXY). KS is the most common chromosomal abnormality associated with male infertility, affecting approximately 1 in every 650 newborn males [18,19]. In addition to the progressive testicular failure and hypergonadotropic hypogonadism associated with infertility, these men have been found to have significantly higher rates of comorbidities and an elevated mortality risk [19]. Specifically, KS men have been found to have higher rates of metabolic syndrome, cardiovascular morbidity, venous thromboembolism, breast cancer, extragonadal germ cell tumors, and reduced bone mineral density [19]. Referring men with infertility to a reproductive urologist creates the opportunity to both diagnose these genetic conditions and counsel these men on the associated health consequences. By identifying these men and referring them to a reproductive urologist, female fertility experts can directly contribute to better disease surveillance, preventative care, and clinical outcomes for these men.

Even in those men without a known cause of infertility, studies have demonstrated an association between reproductive health and overall health. In one European study of 344 men with male factor infertility and 293 age-matched controls, infertile men had a significantly higher rate of comorbidities [20]. After adjusting for age, BMI, and educational status, infertile men still had significantly lower general health versus fertile controls [20]. This finding has been replicated in other larger studies, including a 2015 study by Eisenberg et al., which included over 9300 men evaluated for infertility at an academic center in the United States [21]. They found that the Charlson comorbidity index (CCI) was inversely related to a variety of semen parameters, including semen volume, sperm concentration, total sperm count, and sperm motility (*p*_trend_ < 0.01 for all parameters) [21]. An Italian study also investigated this relationship between semen quality and the CCI, and similarly found that the CCI was inversely associated with sperm concentration (*p* = 0.028) and sperm motility (*p* = 0.06) [22]. Additionally, this group found that increasing the CCI was associated with alterations in reproductive hormones, including higher FSH (*p* = 0.001) and lower total testosterone (*p* = 0.04) [22].

In addition to having lower general health, men with infertility have also been found to be at higher risk for developing certain malignancies and chronic medical conditions. The strongest and most studied relationship between male infertility and malignancy is the correlation between infertility and testis cancer [23]. Several large, retrospective cohort studies performed in the United States have estimated the rate of testicular cancer to be approximately 2 to 3 times higher for men with infertility versus controls, though in smaller series, the risk of testis cancer has been reported to be up to 20 times greater for infertile men with abnormal semen parameters versus controls [24,25,26]. There has also been evidence linking male infertility to prostate cancer [25,27], though this has been challenged in other series [28,29]. Finally, analyses of a large, national insurance database have demonstrated that men with infertility are at higher risk for developing diabetes, ischemic heart disease, and certain autoimmune disorders (including rheumatoid arthritis, multiple sclerosis, psoriasis, thyroiditis, and Grave’s disease) versus controls [30,31].

Given these higher rates of comorbidities, malignancies, and chronic medical conditions, it is no surprise that male infertility has also been associated with increased mortality. In a study of nearly 12,000 men evaluated at two infertility centers in the United States, men with ≥2 abnormal semen parameters were found to have an increased risk of death in both unadjusted (HR 2.96, 95% CI 1.67–5.25) and adjusted (HR 2.29, 95% CI 1.12–4.65) analyses [32]. This risk of mortality seems particularly notable for men with azoospermia. In a 2019 Danish study comparing men who underwent ART versus age-matched controls who conceived naturally, men with azoospermia had an increased risk of death compared to both men who conceived naturally (HR 3.32, 95% CI 2.02–5.40) and the rest of the group that used ART (HR 2.30, 95% CI 1.54–3.41) [33].

While we do not yet know the precise etiology of these associations between infertility, chronic health conditions, and mortality, referral to a reproductive urologist presents an opportunity to identify these pathologies and comorbidities. Ultimately, this initial evaluation by a RU may be essential for optimizing the medical management and overall health of these patients.

### 2.3. Reproductive Urologists Have the Surgical Skills Necessary to Treat Many Causes of Male Infertility and Subfertility

In addition to identifying the pathologies and comorbidities associated with male infertility, a reproductive urologist also possesses the unique surgical skills required to treat male infertility. These skills include various methods for overcoming ejaculatory dysfunction, surgical sperm retrieval techniques, microsurgical varicocelectomy, and microsurgical vasal reconstruction procedures. By collaborating with reproductive endocrinologists, reproductive urologists can use this unique surgical skillset to help couples achieve their family-building goals.

## 3. Surgical Management of Male Infertility

With the recent advances in ART, only a small number of sperm are required for successful oocyte fertilization. Despite these advances, reproductive urologists remain essential for treating couples in which the male partner does not have sperm readily available in their ejaculate, either due to azoospermia or anejaculation. Reproductive urologists can also optimize fertility in nonazoospermic men, thus, allowing couples to use less invasive ART techniques and/or enhancing ART outcomes. Regardless of the underlying pathology, collaboration between male and female reproductive experts is essential for identifying infertile and subfertile men that may benefit from treatment, as well as ensuring that the correct treatment strategy is chosen based on the couple’s unique goals and priorities.

### 3.1. Treatment of Anejaculation and Ejaculatory Duct Obstruction

#### 3.1.1. Electroejaculation

Electroejaculation (EEJ) is a technique that can be used to treat men with anejaculation secondary to a variety of factors, including spinal cord injuries (SCIs), diabetes mellitus, retroperitoneal lymph node dissection, radical pelvic surgery, multiple sclerosis, and psychogenic anejaculation [34,35]. All of these disease processes involve a neurological disruption of the ejaculatory reflex, which arises from the spinal levels T10 to L2 and, subsequently, travels through the sympathetic chain ganglia, hypogastric plexus, and pelvis to the prostate, vas deferens, and seminal vesicles [36]. With this technique, an electrical current is used to induce a neurological response, leading to muscular contraction and the activation of the ejaculatory reflex [37,38].

While a full description of the EEJ technique is beyond the scope of this review, in brief, the procedure begins by catheterizing and fully emptying the bladder [39]. Since retrograde ejaculation is common, a buffering medium and/or human tubal fluid may be instilled into the bladder to preserve any sperm that are deposited there [34,39]. A digital rectal exam and anoscopy are performed to ensure that there are no pre-existing rectal lesions or abnormalities. While men with a complete SCI may undergo the procedure without anesthesia, those with an incomplete SCI or other pathologies typically require general anesthesia (without the use of muscle relaxants) [40]. Blood pressure monitoring is performed throughout the procedure, and those men at risk for autonomic dysreflexia are pretreated with nifedipine [34]. An electrical probe is then inserted into the rectum and positioned with the electrodes in contact with the anterior rectal wall near the prostate and seminal vesicles [34]. Electrical stimulation is administered in progressively increasing increments until ejaculation occurs [34]. The urethra is milked to capture as much antegrade semen as possible, and the bladder is catheterized to capture any retrograde ejaculate. An anoscopy is repeated at the end of the case to ensure that the rectal mucosa has not been injured [34].

By using this technique, sperm can be retrieved up to 90% of the time [41]. This sperm can then be used for intrauterine insemination (IUI) or in vitro fertilization (IVF), with pregnancy rates similar to those of other healthy couples using these ART techniques [34]. In a recent series of over 950 EEJ procedures, the pregnancy rate was 50.0% and live birth rate was 43.8% for couples using sperm obtained with EEJ in combination with in vitro fertilization or an intracytoplasmic sperm injection [40]. No complications due to EEJ were reported [40].

#### 3.1.2. Transurethral Resection of Ejaculatory Ducts (TURED)

Ejaculatory duct obstruction (EDO) is a type of obstructive azoospermia that is present in 1% to 5% of infertile men [42]. For men with EDO, spermatogenesis is typically preserved [42]. Given that sperm production is normal, the current American Urological Association (AUA) and American Society of Reproductive Medicine (ASRM) guidelines state that either a transurethral resection of ejaculatory ducts (TURED) or surgical sperm extraction may be offered as a treatment strategy [43]. Unlike a surgical sperm retrieval procedure, however, a TURED offers couples the chance to conceive naturally, making it an attractive option for those who prefer to avoid invasive and potentially costly ART treatments.

In brief, a TURED procedure is typically performed under general or regional anesthesia, with a surgical setup similar to that used for a transurethral resection of the prostate (TURP) [44]. Specifically, a resectoscope is inserted into the urethra and advanced to the level of the ejaculatory ducts, near the verumontanum [42]. Resection is performed near the verumontanum with an electrocautery loop on a pure cutting current setting to minimize any additional cautery of the ejaculatory ducts, which may result in restenosis [44]. Resection is typically guided with synchronous transrectal ultrasound (TRUS) to confirm the location of the obstruction and avoid iatrogenic rectal injury [44,45]. The resolution of an obstruction is confirmed intraoperatively by the drainage of cloudy, milky fluid from the opened ducts, or by the drainage of methylene blue if transrectal chromotubation of the seminal vesicles was performed [45,46]. While TURED is generally a well-tolerated procedure, complications have been noted in 10% to 20% of patients [47]. These complications primarily include urinary tract infections, epididymitis, hematuria, hematospermia, and watery ejaculate (due to the reflux of urine through widely patent ejaculatory ducts into the seminal vesicles and/or unroofed cysts) [47,48]. There is also a chance of incontinence or rectal perforation given the nature of the procedure, though the risk is low [47].

After a TURED procedure, approximately 60% to 75% of men with EDO demonstrate improvements in semen parameters [46,48,49]. Specifically, the mean ejaculate volume, mean sperm concentration, and mean percent motility have all been found to significantly increase after TURED (*p* < 0.001) [49]. These improvements are both statistically and clinically significant. In one study by Kadioglu et al., nearly three-quarters of the cohort (74%) who underwent TURED demonstrated a >50% increase in postoperative sperm concentration or motility [49]. Additionally, 40% of patients who were previously candidates for IVF or ICSI before surgery (defined as a total motile sperm count ≤ 5 million) were able to achieve a sufficient postoperative total motile sperm count (>5 million) to allow for referral for IUI [49]. In addition to permitting couples to use less invasive ART techniques, spontaneous pregnancy rates after TURED have been found to range between 13% and 30% [46,47,48,49].

Overall, TURED presents another option for some infertile couples affected by EDO to conceive, either naturally or with less invasive ART procedures. By collaborating with reproductive endocrinologists, reproductive urologists are able to play a key role in identifying, diagnosing, and treating these male partners with EDO. Regardless of whether the couple decides to pursue TURED or a surgical sperm retrieval, involving a reproductive urologist in the treatment discussion would ensure that the couple is well-informed about their options and able to determine an educated decision.

### 3.2. Sperm Retrieval Techniques

For those men with normal ejaculatory function and azoospermia, surgical sperm retrieval combined with ICSI offers an opportunity to conceive a biological child. With the advent of ICSI, surgically retrieved sperm from the testis and/or epididymis are able to effectively fertilize oocytes [5]. While the techniques and success rates for treating these men with azoospermia vary significantly depending on the etiology (either obstructive or nonobstructive), a reproductive urologist is a critical part of the reproductive team required to help these couples achieve a pregnancy.

#### 3.2.1. Sperm Retrieval Techniques for Obstructive Azoospermia (OA)

For men with obstructive azoospermia (OA), spermatogenesis within the testis is typically normal. Consequently, sperm retrieval with IVF/ICSI offers a high probability of reproductive success, with sperm retrieval rates reported to be as high as 100% and clinical pregnancy rates of up to 65% [50,51].

For men with an OA secondary congenital bilateral absence of the vas deferens (CBAVD) or other causes not amenable to microsurgical reconstruction, a variety of percutaneous, open, and microsurgical techniques for retrieving sperm from the testis and/or epididymis are available [45]. These techniques include open testicular biopsy (TESE), percutaneous testicular sperm aspiration (TESA), percutaneous testicular biopsy (PercBiopsy), percutaneous epididymal sperm aspiration (PESA), and microsurgical epididymal sperm aspiration (MESA) [45]. While the advantages and disadvantages of each sperm retrieval technique are beyond the scope of this review, it is important to note that MESA has been associated with the best clinical pregnancy rates and large numbers of sperm being retrieved, though microsurgical expertise is required [45]. Regardless of the technique utilized, reproductive urologists play a key role in helping these men with OA conceive biological children—something that would not be possible without the advent of ICSI and continued collaboration between reproductive endocrinologists and reproductive urologists.

#### 3.2.2. Sperm Retrieval Techniques for Nonobstructive Azoospermia (NOA)

In contrast to men with OA, men with nonobstructive azoospermia (NOA) are more difficult to treat due to varying degrees of spermatogenic failure present within the testis [52]. For men with NOA undergoing a surgical sperm retrieval procedure, microdissection testicular sperm extraction (microTESE) is the preferred technique [43].

This procedure involves carefully examining the seminiferous tubules of the testis under an operating microscope at 20–25× magnification to find the focal areas of dilated tubules that are most likely to contain active spermatogenesis [53,54]. By using this technique to identify and selectively remove only dilated tubules, sperm retrieval rates have increased from 16.7–45% (as initially reported with conventional TESE) to as high as 70.8% [55,56,57]. MicroTESE has also been associated with greater numbers of sperm retrieved (160,000 vs. 64,000) and 70-fold less testicular tissue being removed (9.4 mg versus 720 mg) compared to conventional TESE [53,55]. In addition to the benefits to the patient associated with removing only a minimal amount of testicular tissue, this technique also eases the burden on embryology lab personnel. By selecting only the tubules that are most likely to contain sperm, this obviates the need for an extended search through a large volume of tissue and allows laboratory personnel to focus their time and efforts on only the most promising tubules [54].

In addition to higher sperm retrieval rates, microTESE results in lower complication rates, with fewer hematomas, less testicular fibrosis, and less frequent testicular atrophy than TESE [53]. If sperm are retrieved during microTESE and used for ICSI, the average pooled clinical pregnancy rate is 39% [56]; however, clinical pregnancy rates using microTESE sperm have been reported to be as high as 72.4% in some series [57].

While available data from our center and others strongly support the use of microTESE for the treatment of men with NOA, it is important to consider that this is a technically challenging, microsurgical procedure that requires a skilled and experienced surgeon for optimal outcomes [58]. In fact, studies have shown that sperm retrieval rates (SRR) are strongly related to the surgeon’s case volume, with significant improvements in SRR seen after 50 cases and more subtle, continued improvements seen after more than 500 cases [59,60]. This steep learning curve may perhaps be one of the reasons why a recent meta-analysis of 117 studies did not demonstrate any difference in sperm retrieval rates between microTESE and conventional TESE [61]. Ultimately, sufficiently powered and well-designed randomized controlled trials are needed to confirm the superiority of microTESE over conventional TESE. However, given our experience at our center, we believe that by collaborating with reproductive urologists who have advanced microsurgical training and experience performing microTESE procedures, reproductive endocrinologists are able to provide NOA couples with the best chances of conceiving a biological child.

#### 3.2.3. Sperm Retrieval as a Method for Reducing DNA Fragmentation and Enhancing ART Outcomes

While surgical sperm retrieval is an effective method for helping couples with azoospermia conceive, there is also evidence that using testicular and/or epididymal sperm for ICSI may enhance outcomes for couples in which the male partner has an abnormal ejaculated sperm DNA fragmentation (SDF) [62,63,64,65,66,67]. An elevated SDF has been associated with many adverse reproductive outcomes, including lower natural pregnancy rates, lower ART pregnancy rates (including IUI, IVF, and ICSI), abnormal embryo development, and a greater likelihood of recurrent pregnancy loss [68,69,70]. Though many conditions have been associated with an elevated SDF—including environmental factors (i.e., cigarette smoking, radiation, chemotherapy, heat exposure, and medications), pathologic conditions (i.e., varicocele, malignancy, infections, obesity, chronic illness), and even iatrogenic causes (i.e., sperm cryopreservation)—these conditions may lead to DNA damage through similar molecular mechanisms [70,71]. Specifically, these conditions are thought to promote DNA breaks through sperm chromatin packaging defects, apoptosis, and/or oxidative stress [70,71]. While the oocyte may be able to repair some types of sperm DNA damage, this capacity is limited and may vary depending on the individual oocyte [72]. If the damage is not adequately repaired, the embryo cannot develop normally, leading to adverse reproductive outcomes [70,72].

The management of men with elevated SDF presents another opportunity for collaboration between reproductive endocrinologists and reproductive urologists. By identifying couples that have suffered recurrent pregnancy loss or other unexplained infertility, reproductive endocrinologists may be able to identify male partners that are candidates for SDF testing. If abnormal, these men can then be referred to a reproductive urologist for a complete evaluation, including an assessment of risk factors for abnormal SDF (i.e., varicocele, genital tract infections, cigarette smoking, etc.) [68].

For some of these men with elevated SDF, counseling and lifestyle modifications may be beneficial [73]. While AUA/ASRM guidelines concede that there is limited data on the specific lifestyle factors that affect male fertility [17], some studies have demonstrated a positive effect of antioxidant therapy on SDF [74,75,76,77,78,79,80], though this has not been reproduced in all studies [81,82]. Similarly, a short ejaculatory abstinence interval has also been shown to have a positive impact on SDF [83,84]. Finally, given that cigarette smoking [85,86,87,88], air pollution [89,90,91], pesticides [92,93], cancer treatments (including chemotherapy and/or radiation) [94,95], and occupational radiation exposure [96] have all been associated with elevated SDF, it is reasonable to assume that the avoidance of these factors would have a positive impact on SDF, though high-quality data are lacking [73].

Certain men with elevated SDF may also benefit from surgical treatments, such as varicocelectomy or surgical sperm retrieval [73]. While varicocelectomy is discussed in greater detail in the next section, in brief, it has been established that varicocele repair reduces oxidative stress, thus, reducing SDF and contributing to enhanced reproductive outcomes [97]. Similarly, it has been established that sperm retrieved from the testis and/or epididymis has lower levels of SDF [64,65,98]. This is likely because sperm are exposed to oxidative stress during their transit through the male genital tract [99]; by retrieving sperm directly from the testis and/or epididymis, this oxidative stress is avoided, leading to lower SDF levels and better reproductive outcomes using this sperm versus ejaculated sperm [62,68].

Given the invasive nature of a sperm retrieval procedure and current low level of evidence (i.e., no randomized trials) to support using testicular and/or epididymal sperm from nonazoospermic men with elevated SDF, the routine application of this practice remains controversial. The current European Association of Urology (EAU) guidelines recommend approaching this practice with caution, given the risks to the patient associated with invasive procedures [100]. These guidelines clearly state that this technique should only be used when other possible causes of SDF have been excluded, and patients should be counseled on the low-quality evidence available to support this approach [100]. Similarly, AUA/ASRM guidelines note the controversial nature of this practice and limited evidence available to support it; however, they acknowledge that “in a patient with high sperm DNA fragmentation, a clinician may consider using surgically obtained sperm in addition to ICSI” [17]. The most recent European Academy of Andrology (EAA) guidelines may provide the most concrete guidance to clinicians on this topic. The EAA formally recommends that in cases of ≥2 ICSI failures using ejaculated sperm with uncorrectable, elevated SDF, couples should be offered the option of using testicular sperm for ICSI, along with counseling that this approach is based on low-quality evidence [101].

Given the controversial nature of this practice, collaboration between reproductive endocrinologists and reproductive urologists likely presents the best opportunity to identify the couples who would benefit from this procedure. Without clear guidelines or high-level evidence, combining both male and female reproductive expertise is the best way to ensure that couples are receiving the most optimal, evidence-based care for their unique infertility challenges.

### 3.3. Varicocelectomy

Varicoceles are considered to be the most common correctable cause of male infertility [102]. Defined as an abnormal dilation of the pampiniform plexus of the spermatic cord, varicoceles are present in approximately 15% of adult men in the general population, but up to 40% of men with primary infertility and up to 80% of men with secondary infertility [102,103]. A growing body of evidence has identified that varicoceles are associated with negative effects on semen quality, sperm function, reproductive hormone levels, and pregnancy outcomes [102]. While the precise mechanisms by which varicoceles negatively impact male fertility are likely multifactorial and remain under investigation, it is strongly suspected that increased oxidative stress plays a key role [102].

This correlation between varicoceles and oxidative stress is well-established. In 2006, a meta-analysis comparing 118 infertile men with 76 healthy controls found significantly higher reactive oxygen species (ROS) levels (weighted mean difference 0.73; 95% CI 0.40–1.06; *p* < 0.0001) and a lower total antioxidant capacity (TAC) (*p* < 0.00001) in the varicocele group [104]. These elevated ROS levels are likely secondary to multiple factors, including high pressure on venous walls [105], heat stress from scrotal hyperthermia [106,107], hypoxia [106,107], and/or the reflux of renal and adrenal metabolites [102].

Regardless of the etiology, varicoceles have been shown to negatively impact both Sertoli and Leydig cells [108]. On a microscopic level, the seminiferous tubules of men with varicoceles have a thick germinal epithelium, increased apoptosis, and Sertoli cells with extensive cytoplasmic vacuolization [109]. This Sertoli cell dysfunction is observed clinically as a decreased responsiveness to the follicle-stimulating hormone (FSH), decreased androgen-binding protein (ABP), and decreased transferrin levels, all of which contribute to a disruption in spermatogenesis [110]. Similarly, men with varicocele(s) have fewer Leydig cells, and those that are present demonstrate increased cytoplasmic vacuolization and atrophy [111]. Clinically, this is likely responsible for the lower serum testosterone levels observed among men with varicoceles in some studies [112,113,114].

In addition to this negative effect on testicular cell function and spermatogenesis, the hostile biochemical environment created by varicocele(s) may also directly damage sperm. This primary testicular damage may have multiple effects on sperm structure and function, including oocyte-activating factors (such as phospholipase C-zeta), sperm centrosomal components, and sperm DNA integrity. Previous studies have suggested that alterations in these sperm structural and functional components may adversely affect the paternal contribution to final fertilization events and early postfertilization events (i.e., embryonic implantation and development) [115]. While a full description of these postfertilization effects is beyond the scope of this review, there is early evidence to suggest that the primary testicular damage caused by varicoceles may inhibit such embryonic development.

For example, phospholipase C zeta (PLC-z) is a sperm-specific protein that is responsible for oocyte activation after fertilization [116]. After gamete fusion, PLC-z is released into the ooplasm, where it interacts with the oocyte factor(s) to release intracellular calcium ions (Ca^2+^) [117]. These ions regulate a series of molecular events (referred to as ‘oocyte activation’) which are required to initiate embryo development [117]. One study published in 2016 compared 35 men with infertility and varicocele(s) to 20 fertile controls without varicoceles. The authors found that the mean relative expression of PLC-z was significantly lower in men with varicoceles at both the transcriptional and translational levels [118]. While these authors did not provide any additional information on the fertility outcomes of these patients, it has previously been shown that the reduced expression of and/or mutations in PLC-z are associated with low or failed fertilization in infertile men following ICSI [119,120]; thus, it follows that a decrease in PLC-z may be related to the poor IVF/ICSI outcomes seen among men with varicoceles.

Additionally, primary testicular damage to sperm may impact the sperm centrosome, which is required for the nucleation of microtubules and formation of the mitotic spindle [121]. In one study by Hinduja et al., lower centrosome protein expression was found in men with oligoasthenozoospermia compared to normozoospermic men [121]. While these authors did not assess for varicocele, given that varicoceles are known to impair semen parameters, it is possible that such primary testicular damage may affect the sperm centrosome and, subsequently, impair embryo development.

Finally, varicoceles have been found to negatively impact sperm DNA integrity. In one meta-analysis by Wang et al., 240 men with clinical varicoceles had significantly higher levels of sperm DNA damage compared to 176 healthy, fertile controls (mean difference 9.84%; 95% CI 9.19–10.49; *p* < 0.00001) [122]. While the significance of sperm DNA fragmentation (SDF) is still debated, prior studies have found elevated SDF to be associated with lower pregnancy rates, abnormal embryo development, and a greater likelihood of recurrent pregnancy loss [68,69,70]. Fortunately, varicocele repair has been associated with significant improvement in sperm DNA integrity. A meta-analysis published in 2021 analyzed 19 studies and found a significantly lower sperm DNA fragmentation (weighted mean difference −7.23%; 95% CI −8.86 to −5.59; I^2^ = 91%) among men with clinical varicoceles after surgical repair [123].

Ultimately, given the convincing clinical evidence that varicoceles are detrimental to male fertility, recent AUA/ASRM guidelines recommend treating varicoceles in men attempting to conceive who have palpable varicocele(s), infertility, and abnormal semen parameters [43].

#### 3.3.1. Surgical Technique

While a full description of the surgical technique is beyond the scope of this review, it is important to note that a microsurgical approach is considered to be the gold standard, since it has been associated with the highest pregnancy rates, greatest improvements in semen parameters, lowest recurrence rates, and lowest complication rates (versus other nonmicrosurgical techniques) [124,125]. This success is likely due to the enhanced visualization afforded by the operating microscope. By using an operating microscope, the surgeon can see the spermatic cord at up to 25× magnification, which allows for more precise movements, easier identification, the preservation of the testicular arteries and lymphatics, and avoidance of any iatrogenic injuries [125]. Given that this technique requires microsurgical training and expertise, it is important that female infertility specialists collaborate with reproductive urologists to not only identify which patients may benefit the most from this procedure, but also to ensure that they are treated by a provider that is comfortable with a microsurgical technique.

#### 3.3.2. Upgrading Fertility

By treating clinical varicoceles in infertile men, semen parameters may improve significantly enough to allow couples to utilize less invasive forms of ART [103] (Figure 1). This concept of “upgrading” semen quality has been well-described by Samplaski et al. [103]. In this study, the authors evaluated the total motile sperm count (TMSC) of 373 men with varicoceles before and after repair [103]. Overall, TMSC significantly increased from 18.22 ± 38.32 million to 46.72 ± 210.92 million (*p* = 0.007). The authors then defined a TMSC > 9 million as being suitable for natural pregnancy (NP), TMSC 5 million to 9 million as suitable for IUI, and TMSC < 5 million as suitable for IVF. Using this criteria, 58.8% of men initially considered “IVF-only” candidates were upgraded to IUI or NP candidates after varicocelectomy, and 64.9% of men initially considered IUI candidates were upgraded to NP candidates after varicocelectomy [103]. While the authors acknowledge that these TMSC cutoffs are not perfect predictors of conception success, these findings emphasize that varicocele repair may provide couples with the opportunity to use less invasive forms of ART.

In addition to minimizing the burden to the female partner, using less invasive forms of ART is associated with significant cost savings. Given the reported success rates and cost estimates of both varicocelectomy and ICSI, the average cost per live delivery after varicocelectomy is USD 26,268 (95% CI: USD 19,138–USD 44,656) compared to USD 89,091 (95% CI: USD 78,720–USD 99,462) for ICSI [126]. For those couples who pay out-of-pocket for fertility care, this is a significant difference that should be considered by providers.

Ultimately, varicocelectomy presents an opportunity for reproductive urologists and reproductive endocrinologists to collaborate and maximize a couple’s fertility success. This may be of particular importance for those couples who wish to avoid or cannot afford the high costs associated with more invasive forms of ART.

#### 3.3.3. Enhancing IVF Outcomes

Even in those couples that still require IVF/ICSI, varicocelectomy may improve reproductive outcomes. In one recent systematic review and meta-analysis of nonazoospermic infertile men with clinical varicoceles, there was a significant improvement in clinical pregnancy rates (OR = 1.59, 95% CI: 1.19–2.2, I^2^ = 25%) and live birth rates (OR = 2.17, 95% CI: 1.55–3.06, I^2^ = 0%) among men who underwent varicocelectomy prior to ICSI compared to men who proceeded directly to ICSI [127]. Given these enhanced IVF outcomes, varicocelectomy prior to IVF has also been found to be a more cost-effective treatment strategy than proceeding directly to IVF [128]. While the mechanism(s) for this improvement in IVF/ICSI outcomes remains under investigation, it is likely related to the reduction in oxidative stress in seminal plasma and decreased sperm DNA fragmentation associated with varicocelectomy [122,129,130].

Additionally, there is some evidence that varicocelectomy may also be beneficial to men with nonobstructive azoospermia (NOA). While current AUA/ASRM guidelines acknowledge that there is no definitive evidence supporting varicocele repair prior to ART in NOA men, a recent systematic review found an improvement in sperm retrieval rates (SRRs) for men who underwent varicocelectomy prior to sperm retrieval (SRR 48.9% in the treated cohort) versus those who did not (SRR 32.1% in the untreated cohort) [131]. These results are similar to those of a previous systematic review and meta-analysis conducted by Esteves et al., who found a significant increase in SRRs for NOA men with a clinical varicocele who underwent a repair versus those who did not (OR: 2.65, 95% CI 1.69–4.14, *p* < 0.001) [132]. While additional prospective, randomized controlled trials are needed to further evaluate this practice, identifying these patients and referring them to a reproductive urologist for management presents another opportunity for reproductive urologists and reproductive endocrinologists to collaborate. Specifically, it is a chance for a reproductive urologist to optimize the male partner’s fertility and enhance reproductive outcomes: whether that is by increasing the odds that a NOA man may have sperm retrieved for ICSI, or by improving the quality of the ejaculated sperm that is used for IVF/ICSI. In either case, a reproductive urologist is pivotal to the couple’s reproductive success.

#### 3.3.4. Enhancing Testosterone

Finally, in addition to improving reproductive outcomes, treating varicoceles has also been shown to improve testosterone levels. In one meta-analysis of nine studies and 814 men with clinical varicoceles who underwent surgical repair, mean serum testosterone levels improved by 97.48 ng/dL (95% CI 43.73–151.22 ng/dL, *p* = 0.0004) after treatment [133]. To further understand the efficacy of varicocelectomy in treating hypogonadism, a more recent meta-analysis published in 2017 analyzed eight studies (712 men) with subfertility who underwent surgical varicocelectomy [134]. The authors stratified these patients by their preoperative serum testosterone levels, defining “hypogonadal” as a preoperative total testosterone level of <300 ng/dL, and “eugonadal” as a preoperative testosterone level of ≥300 ng/dL. After evaluating all men, the authors found a modest but statistically significant improvement in the mean postoperative total testosterone level of 34.3 ng/dL (95% CI: 22.57–46.04 ng/dL, *p* < 0.0001, I^2^ = 0.0%) [134]. On the subanalysis, however, mean postoperative testosterone levels were significantly greater for hypogonadal men (improved 123 ng/dL, 95% CI: 114.61–131.35 ng/dL, *p* < 0.0001, I^2^ = 37%) compared to eugonadal men and untreated controls [134].

While the precise molecular mechanism(s) behind the negative impact of varicoceles on testosterone production remains to be elucidated [102,135,136], current evidence strongly suggests that varicocele is a risk factors for androgen deficiency [136]. Additionally, given that subfertile men with hypogonadism are more likely to benefit from varicocelectomy, we believe that this cohort should be counseled on and offered surgical repair [134,136]. Since it has long been established that adequate testosterone levels are important for a variety of functions—including libido, erectile function, muscle mass, bone density, and cardiovascular health [137,138,139]—this presents an important opportunity for female fertility specialists to collaborate with reproductive urologists. By identifying men with subfertility, female fertility specialists can help these men receive the appropriate evaluation, testing, and, ultimately, surgical care that they need, which would have a long-lasting, positive impact on their overall health.

### 3.4. Microsurgical Reconstruction

In addition to retrieving sperm for use in IVF/ICSI and helping couples enhance their fertility through varicocele repairs, reproductive urologists also possess the unique technical skills required to treat some patients with obstructive azoospermia through microsurgical reconstruction techniques, including vasovasostomy (VV) and vasoepididymostomy (VE).

Vasovasostomy (VV) involves removing a site of obstruction within the vas deferens and anastomosing the unobstructed abdominal and testicular ends together to restore patency. It is appropriate for patients with vasal obstruction due to a prior vasectomy, iatrogenic vasal injury (i.e., prior inguinal or scrotal surgery), infection, or trauma [45,51]. While a full description of this technique is beyond the scope of this review and more completely described elsewhere [51,140,141], it is important to emphasize that a vasovasostomy is only indicated after an intraoperative vasogram and assessment of vasal fluid, confirming the patency of both the abdominal and testicular ends of the vas deferens [140]. If an abdominal obstruction is noted, there may be a need for additional, advanced surgical maneuvers (such as an inguinal VV or crossed VV) depending on the clinical scenario and patency of the contralateral vas deferens and epididymis [142]. Similarly, if an epididymal obstruction is noted intraoperatively, the surgeon needs to proceed with a VE instead.

A vasoepididymostomy (VE) involves an anastomosis between the vas deferens and an epididymal tubule. Given the size and fragility of the epididymal tubules, experts consider a VE procedure to be considerably more challenging than a VV procedure [140]. As mentioned, this technique is appropriate for patients with an epididymal obstruction, which may be due to longstanding vasal obstruction, trauma, or iatrogenic injury [45,51]. While a full description of the technique is beyond the scope of this review and more completely described elsewhere [140,143,144], it is important to note that the same surgical principles are required for either a successful VV or VE. Namely, both require a high-quality, water-tight, tension-free anastomosis, with close mucosa-to-mucosa approximation and an adequate blood supply [51,140].

While both VV and VE can be successful options for treating infertility in the hands of an experienced microsurgeon, prior studies have consistently demonstrated that VV has a higher success rate than VE. In recent meta-analyses, the pooled mean patency and pregnancy rates for VV were reported to be 89.4% and 73.0% (respectively), versus only 64.1% and 31.1% for VE [145,146]. In certain series, however, patency rates have been reported to be as high as 99.5% for VV [141] and 93% for VE [147].

Achieving these high patency and pregnancy rates requires extensive microsurgical training and expertise. Specifically, hands-on experience is required to master the delicate tissue handling, precise 10-0 suture placement, and intraoperative decision making required for a successful reconstructive procedure [140]. It is unclear exactly how many microsurgical cases a surgeon must perform to overcome this learning curve, though research suggests that providers with a higher surgical volume (≥15 vasectomy reversal cases per year) have better outcomes than those who operate less frequently (<6 cases per year) [148].

For some surgeons, overcoming this learning curve may be a challenge due to their limited exposure to microsurgical cases during residency training. In a recent survey of the Accreditation Council of Graduate Medical Education (ACGME) urology residency programs, 22.4% of programs did not have a fellowship-trained microsurgeon on the faculty [149]. While this survey was unable to assess the microsurgical case volume of these trainees, this finding suggests that approximately one in five United States urology residents may not have exposure to microsurgical training during their residency.

For these residents in particular, microsurgical laboratory training may be essential to compensate for a lack of clinical exposure. In one study by Nagler et al., VV patency rates were 89% for those who practiced in a laboratory versus 53% for those who did not [150]. Another study from Canada similarly found that residents who participated in hands-on VV laboratory training had higher patency rates than those who only received didactic training (54% versus 0%, *p* = 0.01) [151]. Additionally, these authors found that residents who underwent hands-on training retained these skills when tested again 4 months later. Specifically, at this 4-month retention test, the patency rate was 69% for the hands-on group, versus only 20% for the didactic-only group (*p* = 0.05) [151].

These findings emphasize the importance of ensuring that those men who desire a microsurgical reconstructive procedure are referred to an appropriately trained provider. As the number of male infertility fellowships continues to grow, it is likely that the field continues to subspecialize and centers of excellence are likely to emerge [140]. This presents an opportunity for female fertility specialists to identify male partners that desire or may benefit from a reconstructive procedure and refer them to reproductive urologists with the requisite microsurgical training to deliver optimal surgical outcomes. However, there is often debate about the utility of such a reconstructive procedure in the IVF era. This debate is commonly centered around couples in which the male partner has previously undergone a vasectomy.

#### The Role of Vasectomy Reversal (VR) in the IVF Era

While it is considered to be a permanent sterilization procedure, approximately 6% of men undergoing a vasectomy ultimately desire a reversal [152]. For these post-vasectomy patients, both a vasectomy reversal (VR) and sperm retrieval with IVF/ICSI have been found to have similar live birth rates; however, a vasectomy reversal has been shown to be more cost-effective, with the cost per live delivery being less than half that of sperm retrieval with IVF/ICSI [153]. Specifically, Lee et al. reported a cost per live delivery of USD 20,903 for a vasectomy reversal, versus USD 54,797 for a percutaneous testicular sperm extraction (TESE) with IVF/ICSI, and USD 56,861 from microsurgical epididymal sperm aspiration (MESA) with IVF/ICSI [153]. Earlier studies have similarly demonstrated the cost effectiveness of VR compared to sperm retrieval with IVF/ICSI [154,155]—even among some couples with a female partner older than 37 years [156].

Despite this cost-effectiveness, it is important to consider all aspects of the couple prior to recommending a vasal reconstruction procedure versus sperm retrieval with IVF/ICSI. Specifically, female factor infertility, female partner age, obstructed interval, cost of care, and insurance coverage should all be considered when determining which option is the best for a particular couple [157].

For couples affected by female factor infertility, natural conception may be challenging even if the male partner’s vasectomy reversal is successful. While a vasectomy reversal may still be discussed as a potential option, the couple, reproductive urologist, and reproductive endocrinologist should all participate in shared decision making to create a plan that best aligns with the couple’s overall goals [157]. For those couples with such significant female factor infertility that both partners would require reconstructive surgery, the choice to proceed with IVF is clear; in other, less severe pathologies, the discussion may be more nuanced [45].

Similarly, for those couples with an older female partner, the discussion about which treatment strategy to pursue should involve both male and female reproductive experts. At this point, it is well-established that female age is an independent predictor of success after a vasectomy reversal [45]. Specifically, postreversal pregnancy and live birth rates have been found to decrease significantly after ages of 35 to 40 [158,159,160]. However, live birth rates with IVF/ICSI have also been found to drop significantly with maternal age [161,162,163]. Given that female age has a negative impact on both natural conception and IVF/ICSI outcomes, there is no definitive age at which one treatment modality must be pursued over the other [157]. Additional testing (such as ovarian reverse testing) may be helpful in selecting a treatment option when the female partner is at the critical age of 34 to 40 and on the cusp of a precipitous decline in fertility [164]. Additionally, at the time of counseling, it is important to consider not only the female partner’s current age, but the age that she would be once the male has undergone the vasectomy reversal procedure and sperm has returned to the ejaculate. On average, the time to pregnancy after a successful vasectomy reversal is 12 months [165]; this may be considered too long for those women with a smaller window of opportunity to conceive [45]. Clearly, this nuanced conversation is best had in collaboration with both male and female fertility experts, who can guide the couple towards the treatment strategy that is most likely to result in a successful outcome.

While an obstructive interval (OI) has historically been considered an important prognostic factor in VR success [165], more recent studies have demonstrated excellent patency, and at least comparable pregnancy rates can be achieved with prolonged (>10–20 year) obstructive intervals [157,166,167,168]. However, a longer OI has been found to strongly correlate with an increased likelihood of epididymal obstruction and the need for a VE procedure rather than VV [167,168]. Given that VE has a lower likelihood of success than VV, this should be discussed when patients are being counseled on different management strategies.

Finally, the cost of care and insurance coverage should also be considered when counseling a couple on their treatment options. While vasectomy reversal has traditionally been more cost-effective than IVF/ICSI [153,154,155], this may not be the case for couples with insurance coverage for IVF. While 19 states have passed laws that require insurers to either cover or offer coverage for fertility diagnoses and treatment, the qualifications and extent of coverage vary significantly [169]. Providers should consider the laws where they practice and their patient’s insurance coverage options as part of their shared decision-making discussion.

Ultimately, given the many complex male and female fertility factors that contribute to a couple’s decision-making process, the discussion about whether to proceed with a VR or sperm retrieval with IVF/ICSI is another opportunity for reproductive urologists and reproductive endocrinologists to collaborate. By sharing their respective expertise, male and female fertility specialists would be able to provide couples with more comprehensive counseling and enhance their reproductive outcomes.

## 4. Conclusions

Reproductive medicine is a unique field that requires providers to consider the health, goals, and finances of two different individuals to achieve a successful outcome. While recent advances in reproductive technology (particularly ICSI) have allowed for it to be possible to overcome many of the traditional barriers caused by male infertility, it is only through collaboration between both male and female fertility specialists that couples can achieve the optimal reproductive outcomes that are within their unique preferences and possibilities (Table 1). Additionally, through collaboration, providers can provide cost-effective care and have a longstanding positive impact on the health and well-being of infertile men, who we know are at risk for certain comorbidities and chronic conditions.

As our appreciation for and understanding of the complexities of reproductive medicine continue to grow, we are hopeful that male and female reproductive specialists can continue to work together to innovate and treat those impacted by infertility. As we have seen, collaboration is truly more powerful than competition. This is an axiom that we should continue to support in our clinical and research efforts, as well as instill in our trainees.

## Figures and Tables

**Figure 1 jcm-11-04593-f001:**
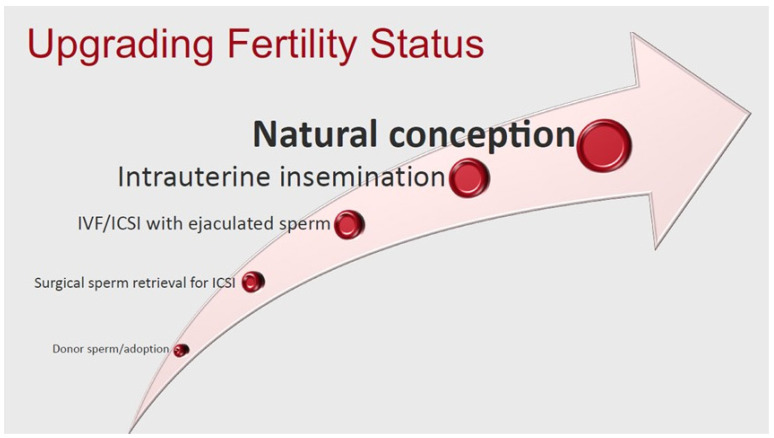
Most couples prefer to conceive naturally. Any treatments that can “upgrade” the fertility status of the couple are beneficial.

**Table 1 jcm-11-04593-t001:** Key points.

Given that modern assisted reproductive technologies (ARTs) can overcome some of the most severe forms of male factor infertility, many men are not referred to a reproductive urologist for a full evaluation. Evaluating the male partner is crucial for optimizing an infertile man’s overall health and providing couples with the least invasive and most cost-effective care. For couples affected by nonobstructive azoospermia (NOA), reproductive urologists are essential for retrieving sperm through advanced microdissection testicular sperm extraction (microTESE) techniques. For couples affected by obstructive azoospermia (OA), reproductive urologists are required to either retrieve sperm or perform a microsurgical vasal reconstruction procedure (vasovasostomy or vasoepididymostomy), which may offer couples the chance to conceive naturally or with less invasive ART techniques. Reproductive urologists can also use their surgical skills to help nonazoospermic couples use less invasive ART techniques and/or optimize ART outcomes through microsurgical varicocelectomy. Using less invasive ART techniques is both cost-effective and may result in improved health outcomes for the offspring, though additional high-quality evidence is needed to fully understand this potential risk. Through collaboration, male and female fertility specialists can combine their relative expertise to help couples successfully navigate the complex, rapidly evolving world of reproductive medicine and contribute to better reproductive outcomes.

## Data Availability

Not applicable.

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
