# Peer review of "Microsurgical Management of Male Infertility: Compelling Evidence That Collaboration with Qualified Male Reproductive Urologists Enhances Assisted Reproductive Technology (ART) Outcomes"

_jcm, 2022, doi:10.3390/jcm11154593_

Round 1

Reviewer 1 Report

This is a good study, and it is well presented. I only perceive some repetitions in the text, and that there are still more studies to have more conclusive results. But it advances in knowledge.

- Lessons learned should be indicted, providing the background context for the paper in the introduction. Please indicate the Aims, objectives of the paper, and the current state of the research.
- Finally, this manuscript was good and saw some grammar or word issues.

Author Response

Point 1: Lessons learned should be indicted, providing the background context for the paper in the introduction. Please indicate the Aims, objectives of the paper, and the current state of the research.

We thank the reviewer for this comment. Given this feedback, we have more clearly stated the aim and objective of the paper in lines 17 and 46-49. Additional language was added in the introduction to clarify the current state of the research  

Point 2: Finally, this manuscript was good and saw some grammar or word issues.

We thank the reviewer for his/her time and thoughtful feedback. We have reviewed the manuscript and corrected any grammar/word issues as appropriate. 

Reviewer 2 Report

The article is well written and easy to read, and the usage of English is correct. With the use of Turnitin software, there was no evidence of plagiarism. This is an excellent manuscript both for physicians and basic scientists. The authors discuss extensively the effects of several testicular pathophysiologies on male reproductive potential.  

However, it may be for the benefit of the reader if the authors touch on the following topics:

1. Post-fertilization effects of primary testicular damage.  In other words, the effects of primary testicular damage on 
a) sperm centrosomal material, 
b) phospholipase Sz (doi: 10.1007/s10815-016-0802-5.
c) DNA integrity 

2. Effects of primary testicular damage either in humans or animals should be discussed

3. A major point is that the effects of left varicocele on Sertoli cell secretory function in humans or animals are not discussed.  

4. Since the authors do not attempt to discuss the topic of c-TESE vs. m-TESE it appears that the metanalysis of Corona et al. 2019 (doi: 10.1093/humupd/dmz028.did not demonstrate a statistically significant SRR after m-TESE should be extensively discussed. Otherwise, the information provided to the reader is not complete. 

I enjoyed reading the manuscript and I congratulate the authors, however, obviously addict studies should be discussed for the benefit of the reader. 

Round 2

Reviewer 2 Report

I enjoyed reading the manuscript and I congratulate the authors.